# Simultaneous alleviation of verification and reference standard biases in a community-based tuberculosis screening study using Bayesian latent class analysis

**Alfred Kipyegon Keter** [1,2,3]*, **Fiona Vanobberghen** [4,5], **Lutgarde Lynen** [1], **Alastair Van Heerden** [2,6], **Jana Fehr** [7,8], **Stephen Olivier** [7], **Emily B. Wong** [7,9], **Tracy R. Glass** [4,5], **Klaus Reither** [4,5], **Els Goetghebeur** [3], **Bart K. M. Jacobs** [1]

**1** Institute of Tropical Medicine, Antwerp, Belgium, **2** Center for Community Based Research, Human Sciences Research Council, Pietermaritzburg, South Africa, **3** Ghent University, Ghent, Belgium, **4** Swiss Tropical and Public Health Institute, Basel, Switzerland, **5** University of Basel, Basel, Switzerland, **6** Faculty of Health Sciences, Department of Paediatrics, SAMRC/WITS Developmental Pathways for Health Research Unit, School of Clinical Medicine, University of the Witwatersrand, Johannesburg, Gauteng, South Africa, **7** Africa Health Research Institute, Durban, South Africa, **8** Hasso-Plattner-Institute for Digital Engineering, Potsdam, Germany, **9** University of Alabama at Birmingham, Birmingham, Alabama, United States of America

* keteralfred@gmail.com

## Abstract

### Background

Estimation of prevalence and diagnostic test accuracy in tuberculosis (TB) prevalence surveys suffer from reference standard and verification biases. The former is attributed to the imperfect reference test used to bacteriologically confirm TB disease. The latter occurs when only the participants screening positive for any TB-compatible symptom or chest X-ray abnormality are selected for bacteriological testing (verification). Bayesian latent class analysis (LCA) alleviates the reference standard bias but suffers verification bias in TB prevalence surveys. This work aims to identify best-practice approaches to simultaneously alleviate the reference standard and verification biases in the estimates of pulmonary TB prevalence and diagnostic test performance in TB prevalence surveys.

### Methods

We performed a secondary analysis of 9869 participants aged ≥15 years from a community-based multimorbidity screening study in a rural district of KwaZulu-Natal, South Africa (Vukuzazi study). Participants were eligible for bacteriological testing using Xpert Ultra and culture if they reported any cardinal TB symptom or had an abnormal chest X-ray finding. We conducted Bayesian LCA in five ways to handle the unverified individuals: (i) complete-case analysis, (ii) analysis assuming the unverified individuals would be negative if bacteriologically tested, (iii) analysis of multiply-imputed datasets with imputation of the missing bacteriological test results for the unverified individuals using multivariate imputation via chained equations (MICE), and simultaneous imputation of the missing bacteriological test

**Data Availability Statement:** Data are available from the Africa Health Research Institute (AHRI) (via https://data.ahri.org/index.php/catalog/990/

study-description#metadata-data_access) for researchers who meet the criteria for access to confidential data. The data underlying the results presented in the study are available from https://data.ahri.org/index.php/catalog/990/study-description#metadata-data_access.

**Funding:** This project is part of the European Union and Developing Countries Clinical Trials Partnership (EDCTP2) programme supported by the European Union (grant number: RIA2018D-2498; TB TRIAGE+). LL, AVH, KR jointly applied for the EDCTP grant. The findings and conclusions of this work are guaranteed by the authors and do not necessarily represent the official position of the funders. The funders had no role in the design, data collection and analysis, decision to publish or preparation of the manuscript.

**Competing interests:** The authors have declared that no competing interests exist.

results in the analysis model assuming the missing bacteriological test results were (iv) missing at random (MAR), and (v) missing not at random (MNAR). We compared the results of (i)-(iii) to the analysis based on a composite reference standard (CRS) of Xpert Ultra and culture. Through simulation with an overall true prevalence of 2.0%, we evaluated the ability of the models to alleviate both biases simultaneously.

## Results

Based on simulation, Bayesian LCA with simultaneous imputation of the missing bacteriological test results under the assumption that the missing data are MAR and MNAR alleviate the reference standard and verification biases. CRS-based analysis and Bayesian LCA assuming the unverified are negative for TB alleviate the biases only when the true overall prevalence is <3.0%. Complete-case analysis produced biased estimates. In the Vukuzazi study, Bayesian LCA with simultaneous imputation of the missing bacteriological test results under the MAR and MNAR assumptions produced overall PTB prevalence of 0.9% (95% Credible Interval (CrI): 0.6–1.9) and 0.7% (95% CrI: 0.5–1.1) respectively alongside realistic estimates of overall diagnostic test sensitivity and specificity with substantially overlapping 95% CrI. The CRS-based analysis and Bayesian LCA assuming the unverified were negative for TB produced 0.7% (95% CrI: 0.5–0.9) and 0.7% (95% CrI: 0.5–1.2) overall PTB prevalence respectively with realistic estimates of overall diagnostic test sensitivity and specificity. Unlike CRS-based analysis, Bayesian LCA of multiply-imputed data using MICE mitigates both biases.

## Conclusion

The findings demonstrate the efficacy of these advanced techniques in alleviating the reference standard and verification biases, enhancing the robustness of community-based screening programs. Imputing missing values as negative for bacteriological tests is plausible under realistic assumptions.

## Introduction

Tuberculosis (TB) prevalence surveys are used to estimate the prevalence of pulmonary TB (PTB) in the investigated population [1]. However, the estimate of PTB prevalence suffers from two types of biases. The first type is the reference standard bias, which occurs when a reference test with imperfect diagnostic performance is used to confirm the presence of TB disease [2]. Many TB diagnostic studies define a TB case as "bacteriologically confirmed" if the specimen yields a positive test result from either sputum smear microscopy (hereinafter, SSM), liquid culture (Mycobacteria Growth Indicator Tube (MGIT); Becton Dickinson, Franklin Lakes, NJ, USA), or a rapid molecular test such as Xpert® MTB/RIF (Cepheid, Sunnyvale, CA, USA) (hereinafter, Xpert) or Xpert® Ultra MTB/RIF (hereinafter, Xpert Ultra) [1,3–5]. In other settings, "bacteriologically confirmed" TB cases were defined using a combination of SSM and culture [6,7], or Xpert Ultra and culture [8,9]. Each of these bacteriological tests has an imperfect sensitivity to detect TB [10–13]. A composite reference standard derived from them is also imperfect and may yield biased estimates of PTB prevalence [2,14].

The second type of bias affecting prevalence estimates is the verification bias. Due to the high cost of bacteriological testing, TB prevalence surveys often prescribe bacteriological testing only when a participant reports at least one TB-compatible symptom or has an abnormal chest X-ray finding [1,4–7]. Asymptomatic individuals with normal chest X-rays are not further tested. Nonetheless, studies have shown that up to 50% of bacteriologically confirmed TB cases in TB prevalence surveys have subclinical TB. Furthermore, chest X-rays alone can miss 2%-27% of active prevalent bacteriologically confirmed TB [15]. In prevalence surveys, verification bias therefore occurs because the probability of obtaining a confirmatory test depends on the results of the previous test(s) and/or covariate(s) related to the disease status [16]. Verification bias can be exacerbated when a subset of the participants eligible for bacteriological testing fails to be tested, e.g. because participants were unable to produce sputum. Suggestions to handle the verification bias in prevalence estimates exist, but are unsatisfactory. One suggestion is to include untested participants as PTB negative, but this may underestimate the true positivity rate. A complete-case analysis is bound to yield biased estimates as well [17]. This is because symptomatic individuals and/or those with abnormal chest X-ray findings have a higher risk for PTB compared to asymptomatic individuals with normal chest X-ray findings. Hence, analyzing the high-risk group alone will yield upward biased estimate of PTB prevalence. This will also yield biased estimates of diagnostic test accuracy. Multiple missing value imputation in combination with inverse-probability weighting has been suggested as the best approach for alleviating verification bias in TB prevalence surveys [17]. Statistical remedies to alleviate the reference standard bias in prevalence do not yet exist. This work aims to identify a best-practice strategy to simultaneously alleviate the reference standard and verification biases in prevalence and diagnostic test accuracy estimates.

To this aim, we employed latent class analysis (LCA), a statistical method that can alleviate the reference standard bias by identifying unobserved subgroups in a population using the test results of all the imperfect diagnostic tests [12]. Besides estimating PTB prevalence, LCA allows the estimation of the sensitivity and specificity of all the imperfect diagnostic tests included in the analysis [12,18,19]. We extended LCA to allow simultaneous alleviation of the reference standard and verification biases in PTB prevalence and diagnostic test sensitivity and specificity estimates. Using a simulated dataset, we evaluated the ability of our method to alleviate both types of biases. We then applied our model to a community-based multi-morbidity screening study (herein referred to as "Vukuzazi" study) conducted in South Africa [8].

## Methods

### Active TB case-finding study in KwaZulu-Natal, South Africa ("Vukuzazi" study)

We conducted a secondary analysis of data collected between May 2018 and May 2019 in a community-based multi-morbidity screening study in the rural uMkhanyakude district of northern KwaZulu-Natal, South Africa. Details of the original study are described elsewhere [8,20]. Briefly, 9914 consenting residents of the catchment area aged ≥15 years were enrolled in the study. Participants were eligible for bacteriological sputum confirmation if they reported at least one of the four cardinal TB symptoms (cough, fever, night sweats and weight loss) or had an abnormal chest X-ray finding. Chest x-rays were analyzed for lung field abnormalities by a radiologist and the computer-aided detection software 'CAD4TB' (Delft Imaging, NL) version 5. CAD4TB scored digital chest X-ray images between 0 to 100 with higher scores suggesting a higher likelihood of PTB due to lung field abnormalities. Participants with CAD4TB scores above the predefined triaging threshold of 25 were referred for bacteriological sputum confirmation. Later, CAD4TB version 6 (CAD4TBv6) and CAD4TBv7, updated versions of

CAD4TBv5, became available at different time points and were used to retrospectively interpret the digital chest X-ray images. A criterion of at least one reported TB-compatible symptom and/or CAD4TBv5 score ≥60 was used between May and September 2018. Bacteriological sputum examination was conducted using Xpert Ultra and culture. Forty-five participants were excluded from the analysis (42 were actively on TB treatment, one had missing TB treatment status and two had missing CAD4TBv7 scores). Of the remaining 9869 participants, 6369 (65%) were eligible for bacteriological testing and 4942 (78%) of those were bacteriologically tested. A consort diagram detailing the eligibility and inclusion of participants in the analysis is presented S1 Fig in Appendix A of the S1 File. Our objective was to estimate the proportion of cases with pulmonary TB (hereinafter, PTB prevalence) and diagnostic test sensitivity and specificity of any TB symptom, radiologist conclusion, CAD4TBv7≥18.28, Xpert Ultra and culture while correcting for the reference standard and verification biases. Only the most recent and updated CAD4TB version 7 was included in the analysis. The cut-off value of 18.28 for CAD4TBv7 was chosen based on the ability to yield a sensitivity of ≥90% when compared to a composite reference standard of Xpert Ultra (excluding trace) and/or culture (S2 Fig in Appendix A of the S1 File). The semiquantitative 'trace' category defined for Xpert Ultra representing a group with a limited amount of detected bacterial load was classified as negative for TB [11,12].

## Notation

Let the random variable $Y_j, j = 1,2,\ldots, J$ denote the result of the $j^{th}$ diagnostic test and the random variable $D$ denote the unknown true PTB status such that $Y_j = 0(1)$ if the $j^{th}$ diagnostic test result is negative (positive) and $D = 0(1)$ if the true PTB status is negative (positive). In this setting, we consider diagnostic tests to include TB-compatible symptoms, radiography-based methods such as radiologist interpretation, and laboratory-based tests such as culture. Further, suppose that the individuals who have positive test results on at least one of the first $t$ diagnostic tests ($t < J$) are verified using the most accurate but expensive test(s). Let the random variable $V = 0(1)$ if an individual is unverified (verified). For unverified participants, the test results for $Y_j, j = t + 1, t + 2,\ldots, J$ are missing. Finally, let the random variable $G$ denote the resulting mutually exclusive subgroups defined by the combination of verification status and eligibility for verification such that $G = 1$ represents those eligible and verified, $G = 2$ those eligible but unverified and $G = 3$ those ineligible and unverified.

## Simulation

We simulated data mimicking the survey design of the Vukuzazi study because the Vukuzazi data had no information on the true diagnostic status of untested individuals [8]. We sequentially generated data for six binary hypothetical diagnostic tests $Y_j, j = 1, 2,\ldots, 6$. In the simulation, $Y_1$ through $Y_6$ play the role of any TB symptom, any chest X-ray abnormality, CAD4TBv5 score, CAD4TBv7 score, Xpert result, and culture result, respectively. Diagnostic tests based on similar biological mechanisms are known to be dependent [21,22]. Thus, $Y_2, Y_3$ and $Y_4$ based on the same chest X-ray images indicating the presence or absence of lung field abnormalities are dependent, separately among the true PTB and non-PTB cases. Similarly, $Y_5$ and $Y_6$ are based on the same sputum specimen, which induces dependence between them only among the true PTB cases. As previously demonstrated elsewhere, [12] we allowed the set $Y_2, Y_3$ and $Y_4$ as well as the pair $Y_5$ and $Y_6$ to be strongly dependent among the true PTB cases, and the set $Y_1, Y_2, Y_3$ and $Y_4$ to have strong dependence among the true non-PTB cases. Only pairwise conditional dependence was considered. Based on the chain rule of conditional probability, we generated the diagnostic test results using regressive logit models such that the joint

probability of $Y = \{Y_1, Y_2, \ldots, Y_6\}$ is expressed as [23]

$$Pr(\boldsymbol{Y}) = \sum_{d=0}^{1} Pr(D = d) \times Pr(Y_1|D = d) \times Pr(Y_2|D = d, Y_1) \times \ldots \times Pr(Y_6|D$$

$$= d, Y_1, Y_2 \ldots, Y_5) \tag{1}$$

Each of $Y_3$ and $Y_4$ was derived from a beta-distributed random variable with scores ranging from 0 to 1. That is, $C_1 \sim Beta(\mu_1 \times \theta_1, (1 - \mu_1) \times \theta_1)$ and $C_2 \sim Beta(\mu_2 \times \theta_2, (1 - \mu_2) \times \theta_2)$, where $\mu_j, \theta_j, j = 1, 2$ are the mean and precision parameters of the beta distributed random variables respectively, $logit(\mu_1) = \beta_0 + \beta_1 Y_1 + \beta_2 Y_2$ and $logit(\mu_2) = \alpha_0 + \alpha_1 Y_1 + \alpha_2 Y_2 + \alpha_3 C_1$. We defined $C_j^* = C_j \times 100, j = 1, 2$ such that $Y_3 = 1$ if $C_1^* \geq 53$ and 0 otherwise, and $Y_4 = 1$ if $C_2^* \geq 15$ and 0 otherwise. The cut-off values were roughly chosen to achieve realistic true values of sensitivity and specificity for $C_1^*$ and $C_2^*$ at the chosen cut-off. We assume that the correct triaging threshold for $C_1^*$ is 25.

The diagnostic test results were generated such that $Y_5$ and $Y_6$ are observed (i.e., individual PTB status is verified using $Y_5$ and $Y_6$) whenever $Y_1 = 1$ or $Y_2 = 1$ or $C_1^* \geq 25$. Otherwise, $Y_5$ and $Y_6$ are missing. For a random subset of the simulated individuals with $25 \leq C_1^* < 60$, representing 3% of all the cases, we set $Y_5$ and $Y_6$ to missing to illustrate instances when a higher threshold $(C_1^* \geq 60)$ was used to define eligibility for testing using $Y_5$ and $Y_6$. We also set another random subset of the simulated individuals with $Y_1 = 1$ or $Y_2 = 1$ or $C_1^* \geq 25$ to have missing data for $Y_5$ and $Y_6$ to represent individuals who fail to be tested with $Y_5$ and $Y_6$ potentially due to refusal or inability to produce sputum for testing using $Y_5$ and $Y_6$. Overall, 15% of the simulated individuals were not tested using $Y_5$ and $Y_6$. The true PTB prevalence was 3.3% among the simulated individuals with observed $Y_5$ and $Y_6$ and 2.7% among the eligible but unverified individuals (i.e., individuals with $Y_1 = 1$ or $Y_2 = 1$ or $25 \leq C_1^* < 60$ and those with $Y_1 = 1$ or $Y_2 = 1$ or $C_1^* \geq 25$ but not tested with $Y_5$ and $Y_6$). The true PTB prevalence among the simulated individuals with $Y_1 = 0$, $Y_2 = 0$ and $C_1^* < 25$ (group not eligible for testing with $Y_5$ and $Y_6$) was simulated to be 0.1%. Thus, we have three groups; eligible for microbiological testing and tested, eligible but not tested and ineligible. We generated a pseudo-random population of 10000 individuals and replicated it 100 times. We did not predefine the true values for the diagnostic tests but chose the parameters of the regressive logit models that would produce strong dependence between the diagnostic tests while yielding realistic true values of sensitivity and specificity for each diagnostic test (S4 Table in Appendix B of the S1 File) [12,24]. We also present the average covariances and correlations of the 100 replicate datasets (S5 Table in Appendix B of the S1 File). The aim of the analysis was to estimate PTB prevalence (overall and in each of the three groups) as well as the overall diagnostic test sensitivity and specificity of $Y_1$, $Y_2$, $Y_4$, $Y_5$ and $Y_6$. In order to match the analysis of the Vukuzazi dataset we omitted $Y_3$.

## Model

Our goal was to estimate overall PTB prevalence and diagnostic test sensitivity and specificity. However, in the presence of incomplete bacteriological testing, models will typically exclude individuals with missing bacteriological test results. Thus, we derive a model that will include the bacteriologically untested individuals under plausible assumptions. We derive our model under the assumption that we only have two groups: those verified (V = 1) and unverified (V = 0). Extension to more than two groups is straight forward.

Under the assumption of conditional independence between the diagnostic tests, the joint probability of a combination of test results from a set of $J$ diagnostic tests $\boldsymbol{Y} = (Y_1, Y_2, \cdots, Y_J)$,

defined as $Pr(\boldsymbol{Y} = \boldsymbol{y})$ is given by

$$\sum_{d=0}^{1} Pr(D = d) \left[ \prod_{j=1}^{t} Pr(Y_j|D = d)Pr(V = v|Y_1, Y_2, \ldots, Y_t, D = d) \times \left\{ \prod_{j=t+1}^{J} \sum_{v=0}^{1} \frac{Pr(Y_j|D = d, \ V = v)Pr(D = d, \ V = v)}{Pr(D = d)} \right\} \right]$$

$$= \sum_{v=0}^{1} \sum_{d=0}^{1} Pr(V = v)Pr(D = d|V = v) \prod_{j=1}^{J} Pr(Y_j|D = d, V = v)$$

(2)

Where

$$Pr(V = v|Y_1, Y_2, \ldots, Y_t, D = d) = \frac{Pr(Y_1, Y_2, \ldots, Y_t|V = v, D = d)Pr(V = v, D = d)}{Pr(Y_1, Y_2, \ldots, Y_t|D = d)Pr(D = d)}$$

For unverified participants ($V = 0$), the test results for $Y_j, j = t + 1, t + 2, \ldots, J$ are not observed hence hypothetical.

We relaxed the assumption of conditional independence in this model to allow modelling of conditional dependence between the diagnostic tests. We also extended the model to include measured covariates (e.g., HIV status, age and sex) known to affect the prevalence and/or diagnostic test sensitivity and specificity as well as verification status such that

$$Pr(\boldsymbol{Y} = \boldsymbol{y}) = \sum_x \sum_{v=0}^{1} \sum_{d=0}^{1} Pr(\boldsymbol{X} = \boldsymbol{x}) \times Pr(V = v|\boldsymbol{X} = \boldsymbol{x}) \times Pr(D = d|\boldsymbol{X} = \boldsymbol{x}, V = v) \times$$

$$\prod_{j=1}^{J} Pr\left(Y_j|D = d, \boldsymbol{X} = \boldsymbol{x}, V = v\right)$$

(3)

This matches the pattern-mixture models for handling non-ignorable missing outcome data [25]. A detailed derivation of the model is presented in the S1 Supplementary Materials.

Fig 1 shows a heuristic model depicting the relationship between the diagnostic tests, true unobserved PTB status, measured covariates, verification status and unobserved sources of dependence among the diagnostic tests used in the Vukuzazi study. This figure was adapted from Keter et al. 2023 and modified to include the verification status and an additional set of unmeasured variables determining the verification status (bacteriological testing using Xpert Ultra and culture), denoted by $V$ and $W$ respectively.

The figure shows unobserved TB bacillary load $\left(U_1^+\right)$, a marker of PTB infection and a source of conditional dependence between microbiological tests, and unmeasured radiological features $\left(U_2^+\right)$ inducing dependence between radiological interpretations and CAD4TB results among the true PTB cases. Other etiologies $\left(U_3^-\right)$ such as bacterial pneumonia and a history of past PTB induces dependence between any TB symptom, radiological interpretations and CAD4TB results among the true non-PTB cases. A detailed discussion of these unmeasured sources of dependencies is provided elsewhere [12].

Conditioning on V induces spurious dependence between any TB symptom, radiologist conclusion and CAD4TBv7 and the microbiological tests via $W$. Association between the measured covariates and microbiological testing cannot be ruled out. Similarly, PTB prevalence and the diagnostic test accuracy varies by subpopulations defined by the measured covariates [12].

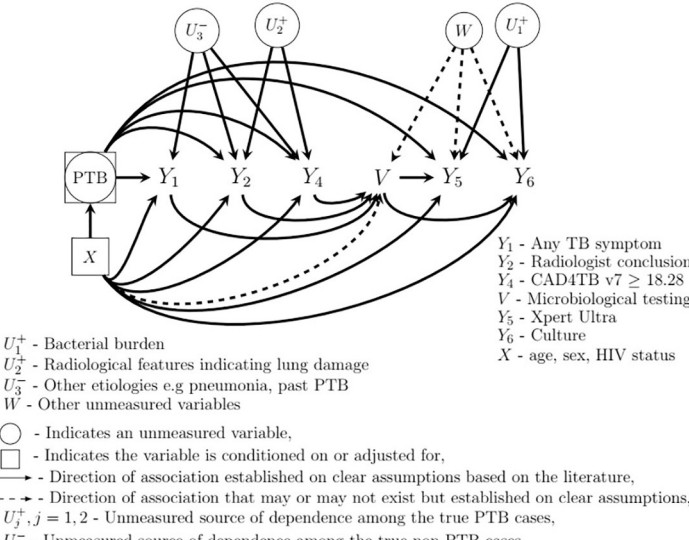

**Fig 1. Heuristic model of diagnostic tests, microbiological testing status, true unobserved PTB status, measured covariates and unobserved sources of dependence between the diagnostic tests used in the Vukuzazi study.**

## Ethics statement

Our study was not separately evaluated by an Ethics committee because it involved the evaluation of statistical methods using simulations and publicly available data, i.e. a methodological rather than a medical research question.

## Statistical analysis

We used two approaches to correct PTB prevalence and diagnostic test accuracy estimates from the reference standard and verification biases: (i) using a composite reference standard (CRS) to estimate PTB prevalence and diagnostic test sensitivity and specificity (CRS-based method) under Bayesian framework, and (ii) Bayesian LCA. CRS-based method assumes conditional independence between the component tests used to derive the CRS and the diagnostic tests under evaluation [2]. The CRS was defined using a combination of $Y_5$ and $Y_6$ using an 'OR' rule that classifies an individual with positive test results on at least one of $Y_5$ and $Y_6$ as disease positive and classifies an individual with negative test results on both $Y_5$ and $Y_6$ as disease negative. No participant had data on one bacteriological test only. Either both were available, or both were missing. In the analysis of Vukuzazi dataset, the Xpert Ultra trace positive only cases are considered as TB negative. This method assumes the CRS is perfect and the estimates of PTB prevalence and diagnostic test accuracy are free from the reference standard bias. Bayesian LCA acknowledges the lack of a perfect reference standard, considers all the diagnostic tests imperfect and incorporates the uncertainties of all the diagnostic tests in a model that classifies individuals into PTB and non-PTB cases and then estimates the diagnostic test sensitivity and specificity. Thus, it should alleviate the reference standard bias. We compared CRS-based method to Bayesian LCA to understand the benefit of Bayesian LCA in alleviating the reference standard bias. For Bayesian LCA, we analyzed both the simulated data and the Vukuzazi dataset using the model that allows conditional dependence between $Y_2$ and $Y_4$ and between $Y_5$ and $Y_6$ among the true PTB cases as well as conditional dependence between $Y_1$, $Y_2$ and $Y_4$ among the true non-PTB cases (as detailed in the Simulation section

above). Using regressive probit models for dependent binary outcomes, we regressed the outcome of one diagnostic test on the unknown PTB status and the preceding diagnostic test(s) as shown in Fig 1 [12].

For each Bayesian approach we conducted (i) complete-case analysis, (ii) analysis assuming the unverified participants were negative and (iii) analysis of 100 multiply-imputed datasets imputing the bacteriological test results of the unverified participants using multivariate imputation via chained equations (MICE) (for the Vukuzazi data only) [26]. We compare the two analyses that included the bacteriologically untested individuals to the complete-case analysis to assess the benefit of the former in alleviating the verification bias. Bayesian LCA should simultaneously alleviate both the reference standard and verification biases while CRS-based analysis should alleviate the verification bias only. Theoretically, analyses that assume the unverified individuals are negative for bacteriological tests will produce incorrect estimates. Imputation of the missing bacteriological test results for the unverified individuals using MICE is based on the assumption that the missing data are missing at random (MAR). This approach fails to acknowledge the dependencies described in the simulation section and depicted in Fig 1. To circumvent these shortcomings, we propose Bayesian LCA with simultaneous imputation of the missing bacteriological test results in the analysis model depicted in Fig 1 under the assumption that the missing data are (iv) MAR, and (v) missing not at random (MNAR). Hence, two additional analyses. This approach runs the analysis and imputes the missing data using the LCA model simultaneously in a single run within the same Markov Chain Monte Carlo (MCMC) algorithm. Hence, this is a distinguishing feature between the analysis of multiply imputed data using MICE and this approach. Given the theoretical limitation of CRS-based analysis in alleviating the reference standard bias, we do not pursue it further. The unknown model parameters were assigned priors from the Gaussian distribution. Except for the unknown parameters of the prevalence model and the regressive probit models for the specificity of Xpert Ultra and culture, the unknown model parameters in all the models were assigned priors from Gaussian distribution with mean zero and unit variance. The unknown model parameters for the overall prevalence and overall false positive rate (equal to 1-specificity) for Xpert Ultra and culture were assigned priors from Gaussian distribution with mean -3 and variance of 0.1. The structure of the models and the priors assigned to the unknown model parameters are presented in S9 and S10 Tables of the S1 File. The choice of the informative prior for the overall prevalence was based on data from the literature on the estimates of population prevalence of PTB in the region where the study was conducted [9]. Similarly, the choice of the informative priors for the overall specificity for Xpert Ultra and culture were based on the literature on the performance of these tests when compared to the composite reference standard and on the expert knowledge. In the absence of TB, culture for *Mycobacterium tuberculosis* has a high specificity [10]. Xpert Ultra also has a substantially high specificity when evaluated against culture [24]. This explains the choice of our priors for the unknown model parameters for the specificity of Xpert Ultra and culture. These analyses are presented in Table 1.

Using Bayesian LCA, we adjusted for the reasons for (non)verification in iii, iv and v. That is, the subgroups defined by eligible and verified (G = 1), eligible but unverified (G = 2) and ineligible and unverified (G = 3). Under the MNAR assumption, the pattern-mixture models for handling non-ignorable missingness in $Y_5$ and $Y_6$ were expressed as $Pr(Y_5 = 1) = \sum_{g\in\{1,2,3\}} \sum_{d=0}^{1} Pr(G = g)Pr(D = d|G)Pr(Y_5 = 1| G = g, D = d)$ and $Pr(Y_6 = 1) = \sum_{g\in\{1,2,3\}} \sum_{d=0}^{1} \sum_{y=0}^{1} Pr(G = g)Pr(D = d|G = g)Pr(Y_6 = 1|G = g, D = d, Y_5 = y_5)$ respectively for $g \in \{1,2,3\}$, $d \in \{0,1\}$ [25].

**Table 1. Approaches used to analyze the simulated and Vukuzazi data.**

| | Simulated data | | Vukuzazi data | |
|---|---|---|---|---|
| Type of analysis | Bayesian LCA | CRS-based method | Bayesian LCA | CRS-based method |
| Complete-case analysis (verified only) | ✓ | ✓ | ✓ | ✓ |
| Verified + unverified assuming the verification test results were negative (i.e., they do not have PTB) | ✓ | ✓ | ✓ | ✓ |
| Verified + unverified with multiple imputation of the missing verification test results using MICE | ‡ | ‡ | ✓ | ✓ |
| Verified + unverified with simultaneous imputation of the missing verification test results assuming that data is MAR | ✓ | X | ✓ | X |
| Verified + unverified with simultaneous imputation of the missing verification test results the data is MNAR | ✓ | X | ✓ | X |

✓-This analysis was conducted,

‡-Not conducted because we did not impute missing data in the simulations

X-Not conducted because we did not carry out simultaneous imputation

For G = 1 we estimated $Pr(Y_5 = 1 \mid G = 1, D = d)$ and $Pr(Y_6 = 1 \mid G = 1, D = d, Y_5 = y_5)$ for $d \in \{0,1\}$ from the observed data. For the participants in G = 2 and 3 we assumed $Pr(Y_5 = 1 \mid G = g, D = d) \leq Pr(Y_5 = 1 \mid G = 1, D = d)$ and $Pr(Y_6 = 1 \mid G = g, D = d, Y_5 = y_5) \leq Pr(Y_6 = 1 \mid G = 1, D = d, Y_5 = y_5)$ where $g \in \{2,3\}$, $d \in \{0,1\}$. Therefore, we estimated $Pr(Y_5 = 1 \mid G = g, D = d) = \emptyset(\epsilon_{5dg})Pr(Y_5 = 1 \mid G = 1, D = d)$ and $Pr(Y_6 = 1 \mid G = g, D = d, Y_5 = y_5) = \emptyset(\epsilon_{6dg})Pr(Y_6 = 1 \mid G = 1, D = d, Y_5 = y_5)$ where $0 \leq \emptyset(\epsilon_{jdg}) \leq 1$, $\epsilon_{jdg} \sim N(0,1)$, $j \in \{5,6\}$, $d \in \{0,1\}$, $g \in \{2,3\}$ and $\emptyset(.)$ is the Gaussian cumulative distribution function [27].

PTB prevalence for the participants in G = 1 was estimated as $Pr(D = 1 \mid G = 1)$. For the participants in G = 2 and 3 we assumed $Pr(D = 1 \mid G = g) \leq Pr(D = 1 \mid G = 1)$, $g \in \{2,3\}$. Therefore, to estimate PTB prevalence in G = 2 and 3 we defined $0 \leq \emptyset(\tau_g) \leq 1$, $\tau_g \sim N(\mu_g, \sigma_g^2)$ for $g \in \{2,3\}$ such that $Pr(D = 1 \mid G = 2) = \emptyset(\tau_2)Pr(D = 1 \mid G = 1)$ and $Pr(D = 1 \mid G = 3) = \emptyset(\tau_3)Pr(D = 1 \mid G = 1)$.

Under the assumption that bacteriological test results were MAR, we defined $\emptyset(\epsilon_{jdg}) = 1$ for $j \in \{5,6\}$, $d \in \{0,1\}$, $g \in \{2,3\}$ so that for G = 2 and 3 $Pr(Y_5 = 1|G = g, D = d) = Pr(Y_5 = 1|G = 1, D = d)$ and $Pr(Y_6 = 1|G = g, D = d, Y_5 = y_5) = Pr(Y_6 = 1|G = 1, D = d, Y_5 = y_5)$. We estimated PTB prevalence as $Pr(D = 1 \mid G = 1) = \emptyset(\omega_1)$, $Pr(D = 1 \mid G = 2) = \emptyset(\omega_2)$ and $Pr(D = 1 \mid G = 3) = \emptyset(\omega_3)$ where $\omega_g \sim N(\mu_g^*, \sigma_g^{*2})$ for $g = 1,2,3$ are priors for the unobserved disease status in each group.

A detailed description of the imputation, analysis of the imputed datasets and the models are presented in Appendix C of the S1 Supplementary Material.

After the analysis of the 100 multiply imputed datasets, we combined the MCMC samples of the posterior distributions from all the 100 analyses as if they were obtained from a single MCMC algorithm. Inferences for the parameters of interest were then based on the resulting posterior distribution. A similar approach was used to summarize the posterior distributions resulting from the analyses of the 100 simulated datasets [28]. We present the estimates and the corresponding 95% credible intervals (95% CrI).

Using MCMC simulation we ran 50000 iterations with the first 25000 discarded as 'warm-up'. For all analyses, convergence in model fitting was assessed by running three chains. Every 10[th] iteration was saved ("thinning") to reduce autocorrelation [29,30]. We used trace plots

and Gelman-Rubin convergence diagnostics to monitor convergence in the chains. All analyses were implemented in R version 4.2.1 using *R2jags* package for R version 4.2.1 [31,32].

## Results

### Simulation

Our simulated data reflected the Vukuzazi dataset, with 46.2%, 15.0% and 36.9% of the participants in the simulation eligible for bacteriological testing and tested, eligible but not tested, and ineligible and not tested, respectively, compared to 49.4%, 14.5% and 35.5% respectively in the Vukuzazi study (S1 Table in Appendix A vs. S6 Table in Appendix B of the S1 File). The simulated data depicted strong dependence between diagnostic tests based on similar biological basis among the true PTB cases and true non-PTB cases (S5 Table in Appendix B of the S1 File). Table 2 presents the results of the analysis of the simulated data using CRS-based analysis and Bayesian LCA.

### CRS-based analysis

Complete-case analysis overestimated the overall prevalence, underestimated the overall sensitivity for $Y_2$ and $Y_4$, and underestimated the overall specificity for $Y_1$, $Y_2$ and $Y_4$. Despite theoretically producing biased estimates, the analysis assuming all the participants with missing

**Table 2. The true values and the posterior median estimates with the corresponding 95% credible intervals (95% CrI) of overall prevalence and overall diagnostic test sensitivity and specificity based on the simulated data.**

| Test | Parameter | True value | CRS-based analysis | | Bayesian LCA | | | |
|---|---|---|---|---|---|---|---|---|
| | | | Complete-case analysis | Analysis assuming the participants with missing values in $Y_5$ and $Y_6$ were negative for $Y_5$ and $Y_6$ | Complete-case analysis | Analysis assuming the participants with missing values in $Y_5$ and $Y_6$ were negative for $Y_5$ and $Y_6$ | Analysis with simultaneous imputation of missing values in $Y_5$ and $Y_6$ assuming the data were MAR | Analysis with simultaneous imputation of missing values in $Y_5$ and $Y_6$ assuming the data were MNAR |
| | | | Median (95% CrI) | Median (95% CrI) | Median (95% CrI) | Median (95% CrI) | Median (95% CrI) | Median (95% CrI) |
| | Prevalence | 2.0 | 4.1 (3.3–4.9) | 2.0 (1.6–2.3) | 3.4 (2.3–5.9) | 2.0 (1.3–3.6) | 1.7 (1.2–2.7) | 1.7 (1.2–2.4) |
| $Y_1$ | Sensitivity | 19.8 | 20.4 (13.1–29.7) | 20.3 (13.0–29.6) | 21.9 (11.9–35.1) | 23.5 (13.9–36.2) | 25.8 (15.9–39.6) | 25.7 (16.6–36.9) |
| | Specificity | 87.4 | 79.1 (77.2–80.7) | 87.4 (86.4–88.3) | 79.1 (77.2–80.7) | 87.4 (86.4–88.4) | 87.6 (86.5–88.6) | 87.6 (86.5–88.5) |
| $Y_2$ | Sensitivity | 66.0 | 57.6 (47.3–67.4) | 57.4 (47.2–67.3) | 67.9 (54.9–81.5) | 68.5 (56.2–82.1) | 63.1 (49.0–74.6) | 63.2 (51.4–73.8) |
| | Specificity | 77.8 | 63.5 (61.4–65.4) | 77.6 (76.5–78.7) | 63.7 (61.6–65.7) | 77.8 (76.6–79.0) | 77.8 (76.6–79.0) | 77.8 (76.6–79.0) |
| $Y_4$ | Sensitivity | 86.9 | 75.0 (66.2–83.0) | 74.9 (66.1–83.0) | 83.2 (58.5–93.0) | 82.8 (56.5–92.2) | 83.5 (73.0–91.4) | 82.9 (73.7–90.4) |
| | Specificity | 70.0 | 54.1 (51.8–56.1) | 69.8 (68.5–71.0) | 54.3 (51.9–56.4) | 70.0 (68.7–71.4) | 70.0 (68.7–71.4) | 70.0 (68.7–71.3) |
| $Y_5$ | Sensitivity | 63.8 | - | - | 66.4 (41.2–86.3) | 60.2 (32.7–77.9) | 67.1 (51.3–82.4) | 67.4 (51.7–82.8) |
| | Specificity | 99.3 | - | - | 99.4 (99.0–99.8) | 99.8 (99.6–99.9) | 99.5 (99.1–99.8) | 99.5 (99.1,99.8) |
| $Y_6$ | Sensitivity | 80.6 | - | - | 85.6 (53.0–99.2) | 76.2 (41.6–94.9) | 83.1 (67.1–94.2) | 83.4 (67.6–94.3) |
| | Specificity | 99.3 | - | - | 99.5 (98.9–99.9) | 99.8 (99.6–100) | 99.5 (98.9–99.9) | 99.4 (98.9–99.9) |

values in $Y_5$ and $Y_6$ were negative for $Y_5$ and $Y_6$, did not show evidence of systematic bias for the overall prevalence and specificity, but underestimated the overall sensitivity for $Y_2$ and $Y_4$. Additional simulations under different assumptions of overall prevalence revealed underestimation of the overall prevalence that became apparent when the true prevalence is $\geq 3.0\%$ (S7 Table in Appendix B of the S1 File).

### Bayesian latent class analysis

Complete-case analysis overestimated the overall prevalence and underestimated the overall specificity for all the diagnostic tests except $Y_5$ and $Y_6$, but showed no evidence of systematic bias for the overall sensitivities. When assuming all participants with missing values in $Y_5$ and $Y_6$ were negative, there was no evidence of systematic bias for the overall prevalence and sensitivity for all the diagnostic tests. Except for $Y_5$ and $Y_6$, this approach also produced estimates of overall specificity for all the diagnostic tests with no evidence of systematic bias. Similarly, under different assumptions of overall prevalence this method underestimated the overall prevalence when the true prevalence is $\geq 3.0\%$ (S7 Table in Appendix B of the S1 File). Lastly, we found no evidence of systematic bias in the estimates of overall prevalence, sensitivity and specificity for all the diagnostic tests based on the analyses that simultaneously imputed the missing values for $Y_5$ and $Y_6$ within the same MCMC algorithm under the assumption that the missing data are MAR or MNAR. These analyses estimated the prevalence to be 3.1% (95% CrI: 2.2–4.3), 1.2% (95% CrI: 0.1–6.7) and 0.0% (95% CrI: 0.0–0.1) among the verified, eligible for verification but unverified, and ineligible participants, respectively, assuming the data were MAR, and correspondingly 3.1% (95% CrI: 2.2–4.3), 1.4% (95% CrI: 0.5–2.6), and 0.1% (95% CrI: 0.0–0.2) assuming the data were MNAR. These results were consistent with the corresponding true values of 3.3%, 2.7% and 0.1%, respectively.

### Vukuzazi study

**Descriptive summary.**   Table 3 shows the distribution of participant characteristics and test results by bacteriological testing status. Additional analysis comparing the eligible and verified participants to those eligible but unverified is presented in the Appendix (S2 Table in Appendix A of the S1 File).

The majority of participants (62.8%) were aged <50 years with only 10.7% aged $\geq 70$ years. Two-thirds of the participants were females and one-third were HIV positive. Visual inspection reveals a higher proportion of individuals aged <50 years and more males were not bacteriologically tested. Of the participants eligible for bacteriological testing, 1427 (22.4%) were not tested. Among the bacteriologically tested participants, 41 (0.8%) were positive for Xpert Ultra and 51 (1.0%) were positive for culture. This translates to 0.4% Xpert Ultra positive and 0.5% culture positive cases among all the participants. Missing covariate data was observed for 52 (0.5%) of all the participants and 15 (0.3%) of the participants who were bacteriologically tested for PTB. For comparison of the estimates across all the analyses, we excluded the participants with missing values in age and HIV status leaving 9817 participants for analysis.

### CRS-based analysis

The assumption of independence between the component tests used to derive the CRS (Xpert Ultra and culture) and the diagnostic tests under evaluation was not violated (S3 Table in Appendix A of the S1 File). As expected, complete-case analysis using CRS approach produced the highest estimate of overall PTB prevalence and the lowest estimates of overall specificity for any TB symptom, any chest X-ray abnormality and CAD4TBv7$\geq$18.28 compared to the other missing data handling methods (Table 4). Complete-case analysis and the analysis

**Table 3. Distribution of participant characteristics and diagnostic test results by verification status.**

| Characteristic | Bacteriologically tested using Xpert Ultra and culture | | Total (n = 9869) |
|---|---|---|---|
| | No (n = 4927; 49.9%) | Yes (n = 4942; 50.1%) | |
| Age (Years) | | | |
| 15–29 | 2322 (47.1%) | 1153 (23.3%) | 3475 (35.2%) |
| 30–49 | 1446 (29.3%) | 1278 (25.9%) | 2724 (27.6%) |
| 50–69 | 892 (18.1%) | 1717 (34.8%) | 2609 (26.4%) |
| ≥70 | 267 (5.4%) | 793 (16.0%) | 1060 (10.7%) |
| Missing | 0 | 1 | 1 |
| Sex | | | |
| Female | 3498 (71.0%) | 3144 (63.6%) | 6642 (67.3%) |
| Male | 1429 (29.0%) | 1798 (36.4%) | 3227 (32.7%) |
| HIV status | | | |
| Negative | 3436 (70.3%) | 3454 (70.1%) | 6890 (70.2%) |
| Positive | 1454 (29.7%) | 1474 (29.9%) | 2928 (29.8%) |
| Missing | 37 | 14 | 51 |
| Any TB symptom[†] | | | |
| No | 4721 (95.8%) | 4103 (83.0%) | 8824 (89.4%) |
| Yes | 206 (4.2%) | 839 (17.0%) | 1045 (10.6%) |
| Chest X-ray lung field findings | | | |
| Normal | 4621 (93.8%) | 3281 (66.4%) | 7902 (80.1%) |
| Abnormal, not suggestive of active TB | 283 (5.7%) | 1495 (30.3%) | 1778 (18.0%) |
| Abnormal, suggestive of active TB | 23 (0.5%) | 166 (3.4%) | 189 (1.9%) |
| CAD4TBv5≥25 | | | |
| No | 3608 (73.2%) | 393 (8.0%) | 4001 (40.5%) |
| Yes | 1319 (26.8%) | 4549 (92.0%) | 5868 (59.5%) |
| CAD4TBv7≥18.28 | | | |
| No | 3680 (74.7%) | 2668 (54.0%) | 6348 (64.3%) |
| Yes | 1247 (25.3%) | 2274 (46.0%) | 3521 (35.7%) |

[†] - A composite of cough and/or fever and/or night sweats and/or weight loss.

assuming the participants not bacteriologically tested were negative on Xpert Ultra and culture produced similar estimates of overall sensitivity. The difference in the estimates of overall sensitivity between the two methods is due to random error in the MCMC sampling. The analysis of multiply imputed data imputing the missing Xpert Ultra and culture test results for all the participants with unconfirmed TB status underestimated the overall sensitivity compared to the other methods. The analysis of multiply-imputed data imputing the missing Xpert Ultra and culture test results for only the eligible but unconfirmed participants produced higher estimates of overall sensitivity but similar estimates of overall specificity for any TB symptom, any chest X-ray abnormality and CAD4TBv7≥18.28 compared to the analysis of multiply-imputed data imputing the missing Xpert Ultra and culture test results for all the participants with unconfirmed TB status.

## Bayesian latent class analysis

Bayesian LCA with complete-case analysis overestimated the overall PTB prevalence while the other methods produced realistic estimates (Table 5). While complete-case analysis produced

**Table 4. Posterior median estimates and the corresponding 95% credible intervals (95% CrI) of overall PTB prevalence and overall diagnostic test sensitivity and specificity based on CRS-based analysis.**

| | | Complete-case analysis | Analysis assuming the participants with missing Xpert Ultra and culture test results were negative for Xpert Ultra and culture | Analysis following multiple imputation of missing Xpert Ultra and culture test results for ALL the participants with unconfirmed TB status ⊥ | Analysis following multiple imputation of missing Xpert Ultra and culture test results for ONLY the eligible participants with unconfirmed TB status ⊥ † |
|---|---|---|---|---|---|
| | | N = 4927 | N = 9817 | N = 9817 | N = 9817 |
| Test | Parameter | Median (95% CrI) | Median (95% CrI) | Median (95% CrI) | Median (95% CrI) |
| | Prevalence | 1.4 (1.1–1.7) | 0.7 (0.5–0.9) | 1.1 (0.9–1.4) | 0.9 (0.7–1.1) |
| Any TB | Sensitivity | 17.7 (10.1–27.8) | 17.6 (9.7–27.7) | 13.7 (7.7–21.8) | 17.3 (9.6–28.0) |
| symptom | Specificity | 83.0 (82.0–84.1) | 89.5 (88.8–90.0) | 89.4 (88.8–90.0) | 89.5 (88.8–90.1) |
| Radiologist | Sensitivity | 82.5 (72.4–90.4) | 82.2 (72.1–89.8) | 59.0 (46.1–71.0) | 74.6 (63.0–84.7) |
| conclusion‡ | Specificity | 67.1 (65.8–68.4) | 80.5 (79.7–81.3) | 80.5 (79.7–81.3) | 80.6 (79.8–81.4) |
| CAD4TBv7 | Sensitivity | 90.1 (81.6–95.7) | 89.9 (88.8–90.0) | 72.1 (59.7–82.8) | 86.0 (75.8–93.2) |
| ≥18.28 | Specificity | 54.6 (53.2–56.0) | 64.7 (63.8–65.7) | 64.8 (63.8–65.7) | 64.8 (63.8–65.8) |

† - Assumes the ineligible participants with missing Xpert Ultra and culture test results were negative for Xpert Ultra and culture

⊥ - Based on 100 multiply imputed datasets.

‡ - Any chest X-ray abnormality.

lower estimates of overall specificity for ay TB symptom, radiologist interpretation and CAD4TBv7≥18.28, the other methods produced higher and similar estimates of overall specificity. Complete-case analysis and the analysis assuming the participants with missing Xpert Ultra and culture test results were negative for Xpert Ultra and culture produced similar estimates of overall sensitivity for all diagnostic tests. The analysis of data with multiple imputation of the missing Xpert Ultra and culture test results for all the participants with unconfirmed TB status produced lower estimates of overall sensitivity for any TB symptom, any chest X-ray abnormality and CAD4TBv7≥18.28 compared to the other methods. Multiple imputation of the missing Xpert Ultra and culture test results for all participants with unconfirmed TB status estimated PTB prevalence to be 1.3% (95% CrI: 0.9–2.1), 0.9 (95% CrI: 0.2–2.6) and 0.1% (95% CrI: 0.0–0.8) among the verified, eligible for verification but unverified and ineligible participants respectively. The analysis with simultaneous imputation of the missing Xpert Ultra and culture test results assuming the missing data is MAR and MNAR produced similar estimates of overall PTB prevalence and overall sensitivity and specificity. However, the analysis assuming the data is MAR produced estimates with wider 95% credible intervals (95% CrI). The analysis assuming the data is MAR estimated PTB prevalence to be 1.5% (95% CrI: 1.1–2.2), 1.7% (95% CrI: 0.3–7.2) and 0.0% (95% CrI: 0.0–0.1) among the verified, eligible for verification but unverified and ineligible participants respectively. The analysis assuming the data is MNAR estimated PTB prevalence to be 1.5% (95% CrI: 1.1–2.1), 0.7% (95% CrI: 0.4–1.3) and 0.1% (95% CrI: 0.0–0.4) among the verified, eligible for verification but unverified and ineligible participants respectively.

We conducted Bayesian LCA with simultaneous imputation of the missing Xpert Ultra and culture test results assuming the data is MAR and MNAR in the Vukuzazi study while

**Table 5. Posterior median estimates and the corresponding 95% credible intervals (95% CrI) of overall PTB prevalence and overall diagnostic test sensitivity and specificity based on Bayesian LCA, unadjusted for measured covariates.**

| Test | | Complete-case analysis | Analysis assuming the participants with missing Xpert Ultra and culture test results were negative for Xpert Ultra and culture | Analysis following multiple imputation of missing Xpert Ultra and culture test results for ALL the participants with unconfirmed TB status ⊥ | Analysis following multiple imputation of missing Xpert Ultra and culture test results for ONLY the eligible participants with unconfirmed TB status ⊥ + | Analysis with simultaneous imputation of missing Xpert Ultra and culture test results assuming the data is MAR | Analysis with simultaneous imputation of missing Xpert Ultra and culture test results assuming the data is MNAR |
|---|---|---|---|---|---|---|---|
| | | N = 4927 | N = 9817 | N = 9817 | N = 9817 | N = 9817 | N = 9817 |
| | | Median (95% CrI) | Median (95% CrI) | Median (95% CrI) | Median (95% CrI) | Median (95% CrI) | Median (95% CrI) |
| | Prev. | 1.3 (0.9–2.2) | 0.7 (0.5–1.2) | 0.8 (0.5–1.5) | 0.7 (0.4–1.4) | 0.9 (0.6–1.9) | 0.7 (0.5–1.1) |
| Any TB | Sn. | 19.1 (10.0–31.0) | 19.2 (10.4–30.7) | 17.2 (8.6–29.0) | 19.2 (9.9–32.2) | 29.1 (11.9–55.5) | 23.0 (12.7–34.6) |
| symptom | Sp. | 83.0 (81.9–84.0) | 89.5 (88.9–90.1) | 89.5 (88.8–90.1) | 89.5 (88.9–90.1) | 89.6 (88.9–90.3) | 89.5 (88.8–90.1) |
| Radiologist | Sn. | 92.1 (81.7–98.1) | 91.8 (81.9–97.8) | 80.4 (58.9–94.0) | 87.4 (71.9–96.8) | 83.8 (53.6–95.1) | 83.6 (71.1–93.4) |
| conclusion‡ | Sp. | 67.3 (65.9–68.7) | 80.6 (79.8–81.5) | 80.6 (79.8–81.5) | 80.7 (79.9–81.5) | 80.7 (79.8–81.6) | 80.6 (79.8–81.4) |
| CAD4TBv7 | Sn. | 96.7 (89.4–99.5) | 96.9 (90.3–99.5) | 90.3 (71.6–98.5) | 94.8 (83.2–99.2) | 88.6 (55.4–97.8) | 89.0 (78.5–96.8) |
| ≥18.28 | Sp. | 54.7 (53.3–56.1) | 64.8 (63.8–65.8) | 64.8 (63.9–65.8) | 64.9 (63.9–65.8) | 64.8 (63.9–65.9) | 64.8 (63.8–65.7) |
| Xpert Ultra† | Sn. | 59.3 (35.4–75.7) | 57.2 (33.8–73.9) | 61.0 (37.1–78.0) | 59.5 (34.6–76.6) | 60.1 (38.9–76.6) | 61.6 (41.3–77.2) |
| | Sp. | 100 (99.9–100) | 100 (99.9–100) | 99.9 (99.8–100) | 100 (99.9–100) | 100 (99.9–100) | 100 (99.9–100) |
| Culture | Sn. | 64.1 (39.7–79.6) | 63.5 (37.9–79.5) | 66.3 (41.6–81.4) | 64.7 (38.7–80.5) | 65.1 (41.7–79.7) | 65.9 (45.4–80.5) |
| | Sp. | 99.8 (99.7–100) | 99.9 (99.9–100) | 99.7 (99.5–99.9) | 99.9 (99.7–100) | 99.8 (99.7–99.9) | 99.8 (99.7–99.9) |

+ - Assumes the ineligible participants with missing Xpert Ultra and culture test results were negative for Xpert Ultra and culture

⊥ - Based on 100 multiply-imputed datasets;

‡ - Any chest X-ray abnormality;

† - Excluding trace; Prev.–Prevalence, Sn.–Sensitivity, Sp. Specificity.

adjusting for age, sex and HIV status (Table 6). Compared to the unadjusted analyses, the adjusted estimates of the overall PTB prevalence and the overall diagnostic test sensitivity and specificity changed slightly. The analyses assuming the data is MAR and MNAR produced similar estimates.

## Discussion and conclusion

In the absence of a perfect reference standard, PTB prevalence and diagnostic test accuracy in TB prevalence surveys are estimated using an imperfect reference test. When the participants bacteriologically tested for TB in the study is not representative of the entire population, the estimates additionally suffer verification bias. We have for the first time proposed an approach to simultaneously alleviate the reference standard and verification biases in PTB prevalence and diagnostic test accuracy in TB prevalence surveys. Estimation of prevalence and diagnostic test accuracy using Bayesian LCA that allowed incorporation of the systematically unverified

**Table 6. Posterior median estimates and the corresponding 95% credible intervals (95% CrI) of overall PTB prevalence and overall diagnostic test sensitivity and specificity based on Bayesian LCA with simultaneous imputation of the missing Xpert Ultra and culture test results in the Vukuzazi study, adjusted for age, sex and HIV status.**

| | | Analysis with simultaneous imputation of missing Xpert Ultra and culture test results assuming the data is MAR | Analysis with simultaneous imputation of missing Xpert Ultra and culture test results assuming the data is MNAR |
|---|---|---|---|
| | | N = 9817 | N = 9817 |
| Test | | Median (95% CrI) | Median (95% CrI) |
| | Prevalence | 0.7 (0.5–1.0) | 0.7 (0.5–1.2) |
| Any TB | Sensitivity | 18.0 (9.9–31.0) | 18.4 (10.0–30.9) |
| symptom | Specificity | 89.3 (88.6–89.9) | 89.3 (88.6–89.9) |
| Radiologist | Sensitivity | 86.1 (73.3–93.7) | 85.3 (71.9–93.4) |
| conclusion[‡] | Specificity | 82.5 (81.7–83.2) | 82.5 (81.8–83.3) |
| CAD4TBv7 | Sensitivity | 87.7 (73.2–95.7) | 87.8 (72.7–95.6) |
| ≥18.28 | Specificity | 65.9 (64.9–66.8) | 65.9 (64.9–66.9) |
| Xpert Ultra[†] | Sensitivity | 61.5 (44.7–77.2) | 61.9 (43.8–77.5) |
| | Specificity | 100 (99.9–100) | 100 (99.9–100) |
| Culture | Sensitivity | 67.3 (53.0–79.2) | 67.4 (51.8–79.6) |
| | Specificity | 99.8 (99.6–99.9) | 99.8 (99.6–99.9) |

[‡] - Any chest X-ray abnormality.

[†] - Excluding trace.

individuals under realistic assumptions alleviated the reference standard and verification biases. Bayesian LCA with simultaneous imputation of the missing bacteriological test results under the assumption that the missing data is MAR and MNAR did not show evidence of systematic bias. Complete-case analysis did not alleviate both biases with CRS-based analysis but alleviated the reference standard bias only with Bayesian LCA. CRS-based analysis and Bayesian LCA assuming the unverified are negative for bacteriological tests alleviate both biases under realistic assumptions.

For the simulated data, we showed that CRS-based analysis and Bayesian LCA assuming all the bacteriologically unverified participants did not have TB alleviated the reference standard and verification biases in the overall prevalence and overall specificity. This method produced plausible estimates of overall PTB prevalence and overall specificities for the Vukuzazi data. Our simulations did not demonstrate the presence of bias in the overall prevalence estimate under the assumption that the unverified individuals are TB-negative. In the simulation, the eligible but unverified comprised 2.7% PTB-positive individuals while the ineligible and unverified comprised 0.1% PTB-positive individuals resulting in a total of 44 PTB-positive individuals. All the unverified individuals including the 44 PTB-positive individuals were assumed not to have TB in the analysis. The findings based on this approach did not reveal any systematic bias but the theoretical estimates of prevalence based on this approach are underestimated. While no evidence of systematic bias in the estimates of the overall sensitivity was observed for some diagnostic tests in the simulated data, this lack of systematic bias may be attributed to the few PTB-positive cases (only an overall prevalence of 2.0%) that result in wider 95% credible intervals that include the true value. Therefore, the lack of evidence of systematic bias in the estimates of overall sensitivity may not be guaranteed in simulations with higher values of true prevalence.

CRS-based analysis of multiply imputed datasets imputing the missing Xpert Ultra and culture test results for all the participants is unreliable. Even the analysis following multiple

imputation of missing Xpert Ultra and culture test results for only the eligible participants produced lower estimates of sensitivity for radiographic methods. The underlying complexities with this approach may be attributed to the MAR assumption conditional on the observed covariates [33]. Therefore, it might have over-imputed false TB cases. On the contrary, Bayesian LCA of the multiply imputed data produced realistic estimates of overall PTB prevalence and diagnostic test sensitivity and specificity. Possibly because Bayesian LCA handles the diagnostic test dependence within the model. Hence, it correctly classified the false positive TB cases leading to realistic estimates.

As the key finding, we highlight that Bayesian LCA with simultaneous imputation of the missing bacteriological test results under the assumption that the data is MAR and MNAR alleviated the reference standard and verification biases in the overall prevalence and overall sensitivity and specificity for all the diagnostic tests in the simulation. In the analysis of the Vukuzazi dataset, Bayesian LCA with simultaneous imputation of the missing Xpert Ultra and culture test results under the assumption that the missing data is MAR and MNAR produced similar and realistic estimates of the overall PTB prevalence (ranging from 0.7% to 0.9%) and diagnostic test sensitivity and specificity with substantially overlapping 95% credible intervals. Except for the complete-case analysis, the other methods produced similar estimates of overall PTB prevalence and overall diagnostic test specificity. The estimated overall PTB prevalence estimates agree with the 0.9% overall estimate from the South Africa National TB prevalence survey [9].

The complete-case analysis of simulated data using the CRS-based approach and Bayesian LCA failed to alleviate the bias in the overall prevalence and specificity. When used to analyze the Vukuzazi data, the complete-case analysis using these two approaches produced implausible estimates of overall PTB prevalence and diagnostic test sensitivity and specificity for any TB symptom, any chest X-ray abnormality and CAD4TBv7 at the chosen threshold score of 18.28. This finding is consistent with the findings reported elsewhere [14].

This study has confirmed for the first time that Bayesian LCA with simultaneous imputation of the missing bacteriological test results under the assumption that the missing data is MNAR can simultaneously alleviate both the reference standard and verification biases when estimating PTB prevalence, and test sensitivity and specificity. Besides the eligible but unverified participants, underestimation in PTB prevalence is contributed by the subclinical TB cases [15,34,35]. Based on our analysis, the ratio of PTB prevalence among the ineligible to the overall shows that up to 15.3% (range: 5.0–36.4) of the true PTB cases could have been missed based on the study design. The uncertainty interval shows that the number of PTB cases in the ineligible and unverified group can be as low as 5% to as high as 36.4% of the total PTB cases. In Kendall et al. 2021, the authors demonstrated that up to 8.7% of bacteriologically confirmed cases were missed because they were asymptomatic with normal chest X-ray findings [34]. Lau et al. 2022 also established that chest X-ray significantly under-detected lung abnormalities and consequently missed up to 3.7% of culture-positive cases [35]. Our findings suggest that in the South African study 13% (based on the estimates of sensitivity of any TB symptom and any chest X-ray abnormality from Bayesian LCA with simultaneous imputation assuming the data is MNAR) of the true PTB cases were asymptomatic with normal chest X-ray findings. This is consistent with the findings by Lau et al. 2022 who showed that up to 17.5% of subclinical cases have normal chest X-ray findings [35]. In light of these evidence, we recommend Bayesian LCA with simultaneous imputation of the missing bacteriological test results assuming the missing data is MNAR for the analysis of diagnostic studies in TB with partial verification to simultaneously alleviate the reference standard and verification biases.

Although Bayesian LCA with simultaneous imputation of the missing Xpert Ultra and culture test results under the assumption that the data is MAR produced plausible estimates, the

assumption that Xpert Ultra and culture have the same diagnostic test sensitivity in the bacteriologically tested and untested participants may be unreasonable. This is the same assumption under which multivariate imputation via chained equations (MICE) is premised. The bacteriologically unverified participants comprised a mix of individuals with different characteristics: some were asymptomatic without lung field abnormalities, others were symptomatic but failed to be microbiologically verified due to inability to produce sputum, failure to be reached at home during follow-up visits or use of a higher CAD4TBv5 triaging threshold [36]. As a consequence, the model underestimated the prevalence of PTB among the asymptomatic participants with normal chest X-ray findings. Therefore, careful consideration of the assumptions should be made prior to applying such a model.

The results based on Bayesian LCA with simultaneous imputation of the missing bacteriological test results assuming the missing data is MAR and MNAR while adjusting for the measured covariates did not seem to alter the conclusions. However, care needs to be exercised to avoid introducing bias via backdoor paths that violate the conditional independence assumption and to avoid overfitting.

We have established that TB symptom screen followed by chest X-ray screening has an imperfect sensitivity. TB screen alone has poor sensitivity in active case-finding while chest X-ray screening alone can miss up to 17% of true TB cases who appear with normal lung fields. Thus, the use of this method as a screening approach needs to be reconsidered. The limitations of the expert radiologist in the interpretation of the digital chest X-ray images can be overcome using CAD4TB that can automatically interpret the images. CAD4TBv7 at a chosen threshold of 18.28 and expert radiologist interpretation of the chest X-ray images have comparable estimates of sensitivity. However, the latter has a lower specificity because CAD4TB interprets any lung field abnormality irrespective of whether it is suggestive of active TB or not. CAD4TB version 7 has low discriminatory power to isolate active TB from other etiologies. Thus, further improvement of the inbuilt artificial intelligence algorithm is required so as to improve on the specificity. Meanwhile, TB diagnostic studies can capitalize on the good sensitivity of expert radiologist interpretations and/or CAD4TBv7 to rule out TB before bacteriological testing. This has the benefit of expanding the population covered with fewer confirmatory tests, and leads to more cases detected [37].

Our findings have revealed that ignoring the uncertainty in the bacteriological tests and the partial information of the unverified participants in the analysis can introduce bias that can be alleviated pragmatically under realistic assumptions. The simple model that simply assumes the unverified individuals are negative for the missing bacteriological test results produce realistic estimates with CRS-based analysis and may be appealing for the analysis of diagnostic studies in TB with partial verification. However, bacteriological tests have imperfect sensitivity for TB detection and the assumption on the true prevalence may be violated. Bayesian LCA with simultaneous imputation of the missing bacteriological test results assuming non ignorable missingness (i.e., MNAR) is robust and safe. Multiple missing value imputation via MICE followed by Bayesian LCA is also feasible.

While our proposed model yields realistic estimates, it cannot be applied directly in a different setting without prior consideration of the diagnostic tests in use, the potential dependencies between the diagnostic tests and the underlying prevalence. The concept is the same but the model will need to be tweaked appropriately. Based on our analysis of active TB case-finding data, regions with low prevalence may require specification of a weakly-informative prior to ensure convergence of the prevalence to a realistic distribution. The factors associated with TB prevalence as well as the diagnostic test properties may be included in the analysis to improve the precision of the estimates.

We have tested our methods using simulation analysis and validated them using only one community-based TB prevalence survey. It would be interesting to understand the performance of the model in several other prevalence surveys. Currently there is an ongoing effort to externally validate these methods in six TB prevalence surveys.

This approach differs from the application in Mungai et al. 2022 who used Bayesian LCA to infer the prevalence of TB among the participants who were not bacteriologically tested while using Xpert and culture as the bacteriological reference standard among the bacteriologically tested participants [38].

This study was not without limitations. Imputation of the missing data can benefit from information from an individual with fully observed data and similar covariates and test results. In our analysis, none of the ineligible and unverified participants was bacteriologically tested. Hence, this might have posed a challenge to the model to learn the outcome of the ineligible participants who had missing bacteriological test results. Potentially, this was also the case with the eligible but unverified participants. Particularly, those who were unable to produce sputum. While we acknowledge that bacteriological testing in TB studies is costly, a random sample of the ineligible participants may need to be bacteriologically tested so as to help with imputation during the analysis. The lack of verification for the eligible participants who could not produce sputum is a sticky problem that needs alternative approaches for TB diagnosis.

The estimates based on our analysis were not weighted by the sampling weights as the model was not developed to account for the sampling weights. Thus, future work can explore the possibility of accounting for the sampling weight.

## Supporting information

**S1 File. The supplementary material is divided into three Appendices; Appendix A, Appendix B and Appendix C.** Appendix A presents additional results based on the analysis of Vukuzazi dataset. Appendix B presents the true values of the simulated data and an additional analysis of the simulated data. Appendix C presents a detailed derivation of the Bayesian latent class model, model structure and the associated parameters as well as the priors. In this appendix, we also present a technical note on how we imputed the missing data in Vukuzazi study using MICE.
(DOCX)

## Author Contributions

**Conceptualization:** Alfred Kipyegon Keter, Lutgarde Lynen, Alastair Van Heerden, Els Goetghebeur, Bart K. M. Jacobs.

**Data curation:** Stephen Olivier, Emily B. Wong.

**Formal analysis:** Alfred Kipyegon Keter.

**Funding acquisition:** Lutgarde Lynen, Alastair Van Heerden, Klaus Reither.

**Methodology:** Alfred Kipyegon Keter, Fiona Vanobberghen, Lutgarde Lynen, Jana Fehr, Tracy R. Glass, Els Goetghebeur, Bart K. M. Jacobs.

**Resources:** Emily B. Wong.

**Software:** Alfred Kipyegon Keter.

**Supervision:** Lutgarde Lynen, Alastair Van Heerden, Els Goetghebeur, Bart K. M. Jacobs.

**Validation:** Jana Fehr, Stephen Olivier, Emily B. Wong, Els Goetghebeur, Bart K. M. Jacobs.

**Writing – original draft:** Alfred Kipyegon Keter.

**Writing – review & editing:** Alfred Kipyegon Keter, Fiona Vanobberghen, Lutgarde Lynen, Alastair Van Heerden, Jana Fehr, Stephen Olivier, Emily B. Wong, Tracy R. Glass, Klaus Reither, Els Goetghebeur, Bart K. M. Jacobs.

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
