## [Decision Letter · Decision Letter 0]

8 Feb 2024

PONE-D-24-00503Simultaneous alleviation of verification and reference standard biases in a community-based tuberculosis screening study using Bayesian latent class analysisPLOS ONE

Dear Dr. Keter, 

Thank you for submitting your manuscript to PLOS ONE. After careful consideration, we feel that it has merit but does not fully meet PLOS ONE’s publication criteria as it currently stands. Therefore, we invite you to submit a revised version of the manuscript that addresses the points raised during the review process.

We look forward to receiving your revised manuscript.

Kind regards,

Novel N Chegou, Ph.D

Academic Editor

PLOS ONE

Journal Requirements:

2. Please be informed that funding information should not appear in the Acknowledgments section or other areas of your manuscript. We will only publish funding information present in the Funding Statement section of the online submission form. Please remove any funding-related text from the manuscript. 

3. For studies involving third-party data, we encourage authors to share any data specific to their analyses that they can legally distribute. PLOS recognizes, however, that authors may be using third-party data they do not have the rights to share. When third-party data cannot be publicly shared, authors must provide all information necessary for interested researchers to apply to gain access to the data. (https://journals.plos.org/plosone/s/data-availability#loc-acceptable-data-access-restrictions) 

**Additional Editor Comments:**

You wrote "NA" under "Ethics statement" in the manuscript submission system. Although this was a secondary analysis of data that was collected via a community-based study, the original study protocol must have been approved by an ethics committee. In the case where only data analysis is done as may be the case here, it is customary in most environments that an ethics committee should still be consulted, for an exemption from ethics certificate if deemed necessary.

Please revise the ethics section in the submission system and add an ethics statement to the main manuscript. Say whether the study was approved (or exemption granted) by an Ethics committee or not.

Reviewers' comments:

Reviewer's Responses to Questions

**Comments to the Author**

1. Is the manuscript technically sound, and do the data support the conclusions?

Reviewer #1: Yes

2. Has the statistical analysis been performed appropriately and rigorously? 

Reviewer #1: Yes

3. Have the authors made all data underlying the findings in their manuscript fully available?

Reviewer #1: Yes

4. Is the manuscript presented in an intelligible fashion and written in standard English?

Reviewer #1: Yes

5. Review Comments to the Author

Reviewer #1: Many thanks for asking me to review this manuscript. Really important methodological study, addressing an issue that we (and many other researchers) have faced when analysing TB prevalence surveys and the accuracy of novel TB diagnostics. Clearly written, and high standard of analysis. I have only a few issues the authors should address:

1. Line 81 and 82: Suggest “higher risk” rather than “high risk”.

2. Fraction of participants in the Vukuzazi study eligible to submit sputum (65%) and prevalence of TB (3.3% in those sputum tested) were higher than in most other national and sub-national prevalence surveys. In the discussion, I would recommend that the authors reflect on whether their results would likely hold when evaluated in settings with less intense TB epidemics.

3. Building upon point 2 above, external validation against other prevalence survey datasets would provide greater confidence in the results of simulations. I don’t think the authors need to do this for the present manuscript, but I wonder if for a future paper, comparison of methods using the SCALE dataset from Malawi (https://journals.plos.org/globalpublichealth/article?id=10.1371/journal.pgph.0001911), the TREATS study from Zambia (https://journals.plos.org/plosmedicine/article?id=10.1371/journal.pmed.1004278), or from recent national TB prevalence surveys might be informative. Either way, I think a sentence in the Discussion should mention external validation.

4. In the methods, would be good to mention priors that were selected for each parameter, and how the influence of prior selection was checked. (I see there is a table of priors in the supplemental material - prior choice can be highly influential in these sorts of models).

5. I note that the authors state in several places that this is the first time this issue has been addressed. Although the authors have done a much more advanced analysis here, we previously undertook a similar approach (albeit without the simulation study) using data from the Kenya National TB prevalence survey (https://journals.plos.org/globalpublichealth/article?id=10.1371/journal.pgph.0001272). Good to reflect on differences in the approach by Mungai et al and the present paper in the discussion.

6. In the supplemental material, trace plots in Figure 3.1 show several chains for parameters with potential anomalies suggestive of serial correlation, likely caused by difficulties exploring the sample sample and low effective samples. It is well known that achieving convergence in LCA models can be challenging. In the supplemental material, the authors should a) describe methods to improve sampling (e.g. potentially related to point 4 above), and b) present a table showing Rhat statistics and effective sample sizes for each parameter.

6. PLOS authors have the option to publish the peer review history of their article (what does this mean?). If published, this will include your full peer review and any attached files.

Reviewer #1: **Yes: **Peter MacPherson

---

## [Author Response · Author response to Decision Letter 0]

8 Apr 2024

Dear Dr. Chegou,

On behalf of the co-authors, I would like to sincerely thank you for allowing our manuscript entitled “Simultaneous alleviation of verification and reference standard biases in a community-based tuberculosis screening study using Bayesian latent class analysis” undergo peer-review. We would also like to thank you and the reviewer for taking time to review our manuscript and for the insightful feedback that has helped us improve the quality of the manuscript.

We wish to resubmit our revised manuscript after considering all the suggestions from the editors and the reviewer. We have particularly revised the manuscript to conform to PLOS ONE’s journal style requirements, have addressed the question on the ethical approval of the study, and responded to the questions and suggestions of the reviewer. These responses are presented with this letter in the next page.

We hope that our responses will be able to sufficiently address the questions raised. Nevertheless, we are happy to respond in case further clarification is required.

Thank you on behalf of the co-authors.

Alfred K. Keter 

Corresponding author

 

Comments to the Author

1. Is the manuscript technically sound, and do the data support the conclusions?

Reviewer #1: Yes

2. Has the statistical analysis been performed appropriately and rigorously?

Reviewer #1: Yes

3. Have the authors made all data underlying the findings in their manuscript fully available?

Reviewer #1: Yes

4. Is the manuscript presented in an intelligible fashion and written in standard English?

Reviewer #1: Yes

5. Review Comments to the Author

Reviewer #1: Many thanks for asking me to review this manuscript. Really important methodological study, addressing an issue that we (and many other researchers) have faced when analysing TB prevalence surveys and the accuracy of novel TB diagnostics. Clearly written, and high standard of analysis. I have only a few issues the authors should address:

1. Line 81 and 82: Suggest “higher risk” rather than “high risk”.

Thank you for this suggestion. We have revised the sentence as suggested.

2. Fraction of participants in the Vukuzazi study eligible to submit sputum (65%) and prevalence of TB (3.3% in those sputum tested) were higher than in most other national and sub-national prevalence surveys. In the discussion, I would recommend that the authors reflect on whether their results would likely hold when evaluated in settings with less intense TB epidemics.

The data presented in the question are from two different sources: fraction eligible for verification - Vukuzazi study, and the prevalence of TB – from the simulation. Nonetheless, we will try to handle this question from two angles. First, the simulation used a TB prevalence of 3.3% among the participants eligible and verified to demonstrate a high-risk group manifesting with symptoms and/or abnormal chest X-ray images. Our goal was to demonstrate the impact of excluding the ineligible and unverified participants from the estimation of prevalence. Analysis of the simulated data yielded the true values used to generate the data. We are confident that if repeated on simulated data using different parameters e.g. lower true values of the prevalence the model would yield correct estimates. In a low prevalence setting, less people would pass the screening step. If a screening test meeting the WHO minimum target product profiles (i.e. sensitivity of 90% and specificity of 70%) is applied on a population setting with 1% prevalence, the resulting estimate of prevalence in the microbiologically tested population is about 3.3%. However, if such tests have a higher specificity in the general population screening (which isn't that unlikely), then a 2-3% prevalence in the microbiologically tested population is still possible with 0.5% prevalence and 80% specificity of the triage test for example. From our analysis of the eligible and verified participants in the Vukuzazi study, we obtained TB prevalence of 1.3% using Bayesian LCA and 1.4% using composite reference standard. Upon accounting for the unverified participants, the estimate based on Bayesian LCA was even lower and closer to the reported overall population prevalence. When adapted appropriately with the use of realistic priors, the model can yield realistic estimates of TB prevalence in regions with less intense TB epidemics. 

As proposed by the reviewer, we have added the following paragraph to the discussion.

“While our proposed model yields realistic estimates, it cannot be applied directly in a different setting without prior consideration of the diagnostic tests in use, the potential dependencies between the diagnostic tests and the underlying prevalence. The concept is the same but the model will need to be tweaked appropriately. Based on our analysis of active TB case-finding data, regions with low prevalence may require specification of a weakly-informative prior to ensure convergence of the prevalence to a realistic distribution. The factors associated with TB prevalence as well as the diagnostic test properties may be included in the analysis to improve the precision of the estimates. “

3. Building upon point 2 above, external validation against other prevalence survey datasets would provide greater confidence in the results of simulations. I don’t think the authors need to do this for the present manuscript, but I wonder if for a future paper, comparison of methods using the SCALE dataset from Malawi (https://journals.plos.org/globalpublichealth/article?id=10.1371/journal.pgph.0001911), the TREATS study from Zambia (https://journals.plos.org/plosmedicine/article?id=10.1371/journal.pmed.1004278), or from recent national TB prevalence surveys might be informative. Either way, I think a sentence in the Discussion should mention external validation.

We wish to thank the reviewer for this suggestion.

Indeed, our methods are currently being applied in the analysis of a number of population-based TB prevalence surveys. Specifically, the methods are being applied in the analysis of ZAMSTAR survey, TREATS study and Bangladesh survey among others. There is an ongoing collaboration between the Institute of Tropical Medicine in Antwerp, Heidelberg University in Germany, Swiss Tropical and Public Health Institute in Switzerland and other partners who are conducting analyses to externally validate these methods in six TB prevalence surveys. We would be glad to collaborate with the reviewer of our manuscript to validate and compare the methods in future work. We have added the paragraph below to the discussion highlighting the ongoing external validations using other datasets.

“We have tested our methods using simulation analysis and validated them using only one community-based TB prevalence survey. It would be interesting to understand the performance of the model in several other prevalence surveys. Currently, there is an ongoing effort to externally validate these methods in six TB prevalence surveys.”

4. In the methods, would be good to mention priors that were selected for each parameter, and how the influence of prior selection was checked. (I see there is a table of priors in the supplemental material - prior choice can be highly influential in these sorts of models).

Thank you for this comment. We have added the paragraph below to the methods section.

The unknown model parameters were assigned priors from the Gaussian distribution. Except for the unknown parameters of the prevalence model and the regressive probit models for the specificity of Xpert Ultra and culture, the unknown model parameters in all the models were assigned priors from Gaussian distribution with mean zero and unit variance. The unknown model parameters for the overall prevalence and overall false positive rate (equal to 1-specificity) for Xpert Ultra and culture were assigned priors from Gaussian distribution with mean -3 and variance of 0.1. The structure of the models and the priors assigned to the unknown model parameters are presented in S9 and S10 Tables of the Supplementary Material. The choice of the informative prior for the overall prevalence was based on data from the literature on the estimates of population prevalence of PTB in the region where the study was conducted. Similarly, the choice of the informative priors for the overall specificity for Xpert Ultra and culture were based on the literature on the performance of these tests when compared to the composite reference standard and on the expert knowledge. In the absence of TB, culture for Mycobacterium tuberculosis has a high specificity. Xpert Ultra also has a substantially high specificity when evaluated against culture. This explains the choice of our priors for the unknown model parameters for the specificity of Xpert Ultra and culture.

5. I note that the authors state in several places that this is the first time this issue has been addressed. Although the authors have done a much more advanced analysis here, we previously undertook a similar approach (albeit without the simulation study) using data from the Kenya National TB prevalence survey (https://journals.plos.org/globalpublichealth/article?id=10.1371/journal.pgph.0001272). Good to reflect on differences in the approach by Mungai et al and the present paper in the discussion.

Thank you for this comment.

Indeed, to the best of our knowledge this is the first application of Bayesian LCA to simultaneously alleviate the reference standard bias as well as the verification bias. Our approach acknowledges that there is no perfect reference standard for conclusive determination of TB. Even culture for Mycobacterium tuberculosis has an imperfect sensitivity. Hence, a negative culture test result does not rule out the presence of TB. 

Using all the available information on the diagnostic tests involved in the diagnosis of TB, without considering any as the reference test, Bayesian LCA was able to stochastically classify the participants into those with and without TB. We extended the model to allow imputation of the unknown TB status among those who were not bacteriologically tested because they were ineligible based on the screening criteria. In doing so, the model was able to yield estimates of TB prevalence and diagnostic test accuracy (sensitivity and specificity) corrected for the verification bias.

This approach differs from the application in Mungai et al. 2022 who used Bayesian LCA to infer the prevalence of TB among the participants who were not bacteriologically tested while using Xpert and culture as the bacteriological reference standard among the bacteriologically tested participants.

We have added the above paragraph into our discussion to highlight the difference between these two approaches.

6. In the supplemental material, trace plots in Figure 3.1 show several chains for parameters with potential anomalies suggestive of serial correlation, likely caused by difficulties exploring the sample sample and low effective samples. It is well known that achieving convergence in LCA models can be challenging. In the supplemental material, the authors should a) describe methods to improve sampling (e.g. potentially related to point 4 above), and b) present a table showing Rhat statistics and effective sample sizes for each parameter.

Thank you for this comment.

We acknowledge that convergence can be problematic with Bayesian LCA. This limitation can be overcome by carefully choosing priors for the unknown model parameters. Informative priors for certain parameters can be chosen based on the expert knowledge and the available data in the literature. This can improve model convergence. Increasing the number of iterations and choosing the correct “burn-in” or “warm-up” period in the MCMC sampling algorithm can also help achieve good mixing. Using a wider sampling interval can reduce autocorrelation. 

Nevertheless, the trace plots presented in S3 Fig. in the Supplementary Material (Figure 3.1 in the previous version of the Supplementary Material) depict convergence in the imputation models used to impute the missing Xpert Ultra and culture test results as well as the HIV status among the microbiologically untested participants. This imputation was implemented using Multivariate Imputation via Chained Equations (MICE) in R statistical software. Except for a few instances where sampling in one of the chains appears a little distant from the rest of the chains, the sampling seems to converge to some value for each of the three presented variables. The reason for the conspicuous anomalies may be explained by the choice of the axis scale which is determined inside the mice package. In the package, 5-10 iterations are recommended but, in our case, we used 50 iterations which provides a better exploration of the sampling space.

6. PLOS authors have the option to publish the peer review history of their article (what does this mean?). If published, this will include your full peer review and any attached files.

We consent to publication of the peer review history of our article.

Do you want your identity to be public for this peer review? For information about this choice, including consent withdrawal, please see our Privacy Policy.

Reviewer #1: Yes: Peter MacPherson

 

Journal Requirements:

Thank you for this comment. We have formatted the manuscript and Supplementary Material to conform to the PLOS ONE’s style requirements.

2. Please be informed that funding information should not appear in the Acknowledgments section or other areas of your manuscript. We will only publish funding information present in the Funding Statement section of the online submission form. Please remove any funding-related text from the manuscript. 

Thanks for this comment. We have removed the funding information from the body of the manuscript. We will ensure that this information is only available in the Funding Statement section of the online submission form.

3. For studies involving third-par

---

## [Decision Letter · Decision Letter 1]

23 Apr 2024

PONE-D-24-00503R1Simultaneous alleviation of verification and reference standard biases in a community-based tuberculosis screening study using Bayesian latent class analysisPLOS ONE

Dear Dr. Keter,

Thank you for submitting your manuscript to PLOS ONE. After careful consideration, we feel that it has merit but does not fully meet PLOS ONE’s publication criteria as it currently stands. Therefore, we invite you to submit a revised version of the manuscript that addresses the points raised during the review process.

**ACADEMIC EDITOR:**Please address the outstanding important query raised by the reviewer. 

 Please submit your revised manuscript by Jun 07 2024 11:59PM. If you will need more time than this to complete your revisions, please reply to this message or contact the journal office at plosone@plos.org. Please include the following items when submitting your revised manuscript:A rebuttal letter that responds to each point raised by the academic editor and reviewer(s). You should upload this letter as a separate file labeled 'Response to Reviewers'.A marked-up copy of your manuscript that highlights changes made to the original version. You should upload this as a separate file labeled 'Revised Manuscript with Track Changes'.An unmarked version of your revised paper without tracked changes. You should upload this as a separate file labeled 'Manuscript'.If applicable, we recommend that you deposit your laboratory protocols in protocols.io to enhance the reproducibility of your results. Protocols.io assigns your protocol its own identifier (DOI) so that it can be cited independently in the future. For instructions see: https://journals.plos.org/plosone/s/submission-guidelines#loc-laboratory-protocols. Additionally, PLOS ONE offers an option for publishing peer-reviewed Lab Protocol articles, which describe protocols hosted on protocols.io. Read more information on sharing protocols at https://plos.org/protocols?utm_medium=editorial-email&utm_source=authorletters&utm_campaign=protocols.

We look forward to receiving your revised manuscript.

Kind regards,

Novel N. Chegou, Ph.D

Academic Editor

PLOS ONE

Journal Requirements:

Reviewers' comments:

Reviewer's Responses to Questions

**Comments to the Author**

1. If the authors have adequately addressed your comments raised in a previous round of review and you feel that this manuscript is now acceptable for publication, you may indicate that here to bypass the “Comments to the Author” section, enter your conflict of interest statement in the “Confidential to Editor” section, and submit your "Accept" recommendation.

Reviewer #1: (No Response)

2. Is the manuscript technically sound, and do the data support the conclusions?

Reviewer #1: Yes

3. Has the statistical analysis been performed appropriately and rigorously? 

Reviewer #1: Yes

4. Have the authors made all data underlying the findings in their manuscript fully available?

Reviewer #1: Yes

5. Is the manuscript presented in an intelligible fashion and written in standard English?

Reviewer #1: Yes

6. Review Comments to the Author

Reviewer #1: Thank you for asking me to review this revised manuscript. The authors have done a good job at responding to all comments.

There is only one issue outstanding that they did not address. To confirm convergence and appropriate exploration of the parameter space, the authors should add a table to the supplemental material showing the Rhat statistic and effective sample size for each model parameter.

7. PLOS authors have the option to publish the peer review history of their article (what does this mean?). If published, this will include your full peer review and any attached files.

Reviewer #1: **Yes: **Peter MacPherson

---

## [Author Response · Author response to Decision Letter 1]

21 May 2024

Date: Apr 23 2024 04:01PM

To: "Alfred Kipyegon Keter" keteralfred@gmail.com

cc: novel@sun.ac.za

From: "PLOS ONE" plosone@plos.org

Subject: PLOS ONE Decision: Revision required [PONE-D-24-00503R1]

PONE-D-24-00503R1

Simultaneous alleviation of verification and reference standard biases in a community-based tuberculosis screening study using Bayesian latent class analysis

PLOS ONE

Dear Dr. Keter,

Thank you for submitting your manuscript to PLOS ONE. After careful consideration, we feel that it has merit but does not fully meet PLOS ONE’s publication criteria as it currently stands. Therefore, we invite you to submit a revised version of the manuscript that addresses the points raised during the review process.

ACADEMIC EDITOR:

Please address the outstanding important query raised by the reviewer. 

We look forward to receiving your revised manuscript.

Kind regards,

Novel N. Chegou, Ph.D

Academic Editor

PLOS ONE

Journal Requirements:

Thank you for this comment.

We apologize that this requirement was not well addressed previously. We have replaced the retracted articles from the reference list as well as from the citations in the article with the correct ones. We hope this action addresses the concerns raised by the editor regarding the cited papers that have been retracted.

Reviewers' comments:

Reviewer's Responses to Questions

Comments to the Author

1. If the authors have adequately addressed your comments raised in a previous round of review and you feel that this manuscript is now acceptable for publication, you may indicate that here to bypass the “Comments to the Author” section, enter your conflict of interest statement in the “Confidential to Editor” section, and submit your "Accept" recommendation.

Reviewer #1: (No Response)

2. Is the manuscript technically sound, and do the data support the conclusions?

Reviewer #1: Yes

3. Has the statistical analysis been performed appropriately and rigorously?

Reviewer #1: Yes

4. Have the authors made all data underlying the findings in their manuscript fully available?

Reviewer #1: Yes

5. Is the manuscript presented in an intelligible fashion and written in standard English?

Reviewer #1: Yes

6. Review Comments to the Author

Reviewer #1: Thank you for asking me to review this revised manuscript. The authors have done a good job at responding to all comments.

There is only one issue outstanding that they did not address. To confirm convergence and appropriate exploration of the parameter space, the authors should add a table to the supplemental material showing the Rhat statistic and effective sample size for each model parameter.

We regret that this request was missed in our previous responses. In the supplementary material, we have added tables of parameter estimates that include the indicators of model convergence (R hat statistic and effective sample size) for the models we fitted. We hope this addresses the question raised by the reviewer.

7. PLOS authors have the option to publish the peer review history of their article (what does this mean?). If published, this will include your full peer review and any attached files.

Do you want your identity to be public for this peer review? For information about this choice, including consent withdrawal, please see our Privacy Policy.

Reviewer #1: Yes: Peter MacPherson

---

## [Decision Letter · Decision Letter 2]

24 May 2024

Simultaneous alleviation of verification and reference standard biases in a community-based tuberculosis screening study using Bayesian latent class analysis

PONE-D-24-00503R2

Dear Dr. Keter,

We’re pleased to inform you that your manuscript has been judged scientifically suitable for publication and will be formally accepted for publication once it meets all outstanding technical requirements.

Kind regards,

Novel Njweipi Chegou, Ph.D

Academic Editor

PLOS ONE

Additional Editor Comments (optional):

Reviewers' comments:

Reviewer's Responses to Questions

**Comments to the Author**

1. If the authors have adequately addressed your comments raised in a previous round of review and you feel that this manuscript is now acceptable for publication, you may indicate that here to bypass the “Comments to the Author” section, enter your conflict of interest statement in the “Confidential to Editor” section, and submit your "Accept" recommendation.

Reviewer #1: All comments have been addressed

2. Is the manuscript technically sound, and do the data support the conclusions?

Reviewer #1: Yes

3. Has the statistical analysis been performed appropriately and rigorously? 

Reviewer #1: Yes

4. Have the authors made all data underlying the findings in their manuscript fully available?

Reviewer #1: Yes

5. Is the manuscript presented in an intelligible fashion and written in standard English?

Reviewer #1: Yes

6. Review Comments to the Author

Reviewer #1: Many thanks for addressing the remaining outstanding issue. Model diagnostics look good - this is a great paper, and I am sure will be high importance in the future.

7. PLOS authors have the option to publish the peer review history of their article (what does this mean?). If published, this will include your full peer review and any attached files.

Reviewer #1: **Yes: **Peter MacPherson

---

## [Editor Report · Acceptance letter]

30 May 2024

PONE-D-24-00503R2 

PLOS ONE

Dear Dr. Keter, 

I'm pleased to inform you that your manuscript has been deemed suitable for publication in PLOS ONE. Congratulations! Your manuscript is now being handed over to our production team.

Kind regards, 

on behalf of

Prof Novel Njweipi Chegou 

Academic Editor

PLOS ONE